# Can Glyphosate and Temperature Affect the Nutritional Lipid Quality in the Mussel *Mytilus galloprovincialis*?

**DOI:** 10.3390/foods12081595

**Published:** 2023-04-09

**Authors:** Francesca Biandolino, Ermelinda Prato, Asia Grattagliano, Isabella Parlapiano

**Affiliations:** 1National Research Council, Water Research Institute (CNR-IRSA), Via Roma, 3, 74123 Taranto, Italy; 2Department of Chemical Sciences and Technologies, University of Rome Tor Vergata, Via Della Ricerca Scientifica, 1, 00133 Roma, Italy

**Keywords:** mussels, Mediterranean Sea, glyphosate, temperature, fatty acids, nutritional quality indices

## Abstract

Mussels are an important source of the essential omega-3 polyunsaturated fatty acids (n-3 PUFAs), which play a critical role in human health, preventing a variety of diseases. This study aimed to evaluate, for the first time, the combined effect of glyphosate (Gly) and culturing temperature on the lipid content and fatty acid (FA) profile of the Mediterranean mussel *M. galloprovincialis*. In addition, a number of lipid nutritional quality indices (LNQIs) were applied as important tools to assess the nutritional value of food. Mussels were exposed for 4 days to two Gly concentrations (1 mg/L, 10 mg/L) and two temperatures (T°: 20–26 °C). Statistical analysis showed significant effects of T°C, Gly, and T°C × Gly interaction (*p* < 0.05) on the lipid and FA profiles of *M. galloprovincialis*. Mussels exposed to 10 mg/L Gly at 20 °C showed a decrease in eicosapentaenoic (EPA, from 14.6% to 12% of total FAs) and docosahexaenoic acids (DHA, from 10% to 6.4% of total FAs), compared to the control mussels. Both stressors caused a considerable decrease in n-3 PUFAs, which resulted in a less favorable n-6/n-3 PUFA ratio. Overall, this study demonstrated a decline in the nutritive values of mussels, most prominently in groups exposed to 10 mg/L Gly at a temperature of 20 °C and in those exposed to a temperature of 26 °C. This was confirmed by such LNQIs as EPA + DHA, PUFA/Saturated FAs, atherogenic and thrombogenic indices (AI and TI), the health promoting index (HPI), and the unsaturation index (UI). Further investigations into chronic exposure to both stressors are desirable to predict the impacts on aquatic ecosystems and food quality.

## 1. Introduction

Marine ecosystems are a rich source of nutritious food needed for human health and wellbeing. Seafood provides a range of key nutrients that are frequently scarce in habitual diets, which is the reason many governments and nutritional guidelines recommend an increase in its consumption. 

Among seafood, mussels play an important role in the human diet, being largely appreciated by consumers for both their organoleptic and nutritional properties. Indeed, they are a good source of high-quality proteins, omega-3 polyunsaturated fatty acids (n-3 PUFAs), and other nutrients [1]. 

The Mediterranean mussel (*Mytilus galloprovincialis*) is the most common species in the European Union, representing 61% of the total mussel production [2]. This species is farmed in the Mediterranean countries (Italy, Greece, France, Spain, Bulgaria, Croatia, Slovenia) as well as in Galicia (Atlantic coast of Spain). Mussel flesh is well-known for its nourishing value, resulting mainly from the presence of n-3 PUFAs such as eicosapentaenoic acid (EPA; C20:5 n-3) and docosahexaenoic acid (DHA, C22:6 n-3). Several studies report that PUFAs contribute most to the total FA profile of *M. galloprovincialis*, in the range of 38–67% of total FAs [3,4,5,6,7,8]. This is due to the diet habit of mussels that, as they are a filter-feeding animal, consists mainly of microalgae (e.g., diatoms, flagellates) [9], which are recognized as the main source of n-3 PUFAs in the marine environment [10]. Therefore, the consumption of mussels as human food represents an environmentally sustainable strategy to improve n-3 PUFA status. 

Fatty acids (FAs) are among the most important natural, biologically active compounds, and the FA profiles of food products has become a topic of great interest for consumers, who are increasingly attentive to adopting a healthier diet. n-3 PUFAs, EPA and DHA, are ubiquitous components of cell membranes and are important for many metabolic and physiological functions. 

Many epidemiological and experimental studies have reported all the important metabolic functions that these FAs facilitate and the health benefits associated with their intake, which make them a functional food. These include normal growth and brain development, protective action against cardiovascular diseases and various inflammatory conditions such as rheumatoid arthritis, psoriasis, asthma, obesity, and some cancers [11,12,13]. There is also increasing evidence that demonstrates that diets containing EPA and DHA may protect against the development of Alzheimer’s disease [14].

n-3 PUFAs are known to be “essential” to human health as they cannot be synthesized by humans. Biochemical pathways to convert the precursor alpha-linolenic acid (ALA, 18:3-n-3) to EPA and, further, to DHA exist, but the conversion rates in the human body are low; about 8% and 4% of ALA in healthy adults is converted to EPA and DHA, respectively [15,16,17,18,19]. The International Society for the Study of Fatty Acids and Lipids (ISSFAL) reported the conversion efficiency of ALA to DHA in the order of 1% in infants, and lower in adults [19].

Therefore, consumption of EPA and DHA directly from marine food sources is the most efficacious method to increase the levels of these PUFAs in the body. In addition, mussels are an environmentally sustainable way of producing dietary protein, PUFAs, and n-3 fatty acids, as they result in lower greenhouse gas emissions compared to the production chains of other animals [1,20].

Since mussels demonstrate extensive filter-feeding activity, they can also retain and concentrate pollutants from the marine environment in their body, increasing the risk to human health. Among them, the most dangerous pollutants that can be found in mussels are heavy metals, polycyclic aromatic hydrocarbons, pesticides, and pathogenic bacteria [21,22,23,24,25,26,27].

In most high-income countries, agricultural activities represent the principal factor in the degradation of coastal waters, mainly due to the massive use of plant protection products. Among them, glyphosate-based herbicide (GBH) is one of the most widely used globally, with an increase of more than 12 times to 826 million kg having occurred between 1995 and 2014 [28]. Glyphosate (N-(phosphonomethyl)Glycine) is a broad-spectrum herbicide used for weed control in agriculture. The suggested mode of action of glyphosate is the inhibition of the key enzyme involved in aromatic aminoacid synthesis in plants and many microorganisms, compromising protein synthesis and growth, which ultimately leads to cellular disarray and death [29]. Glyphosate (Gly) is highly soluble in water, and it has been detected in different aquatic ecosystems adjacent to intensive agricultural areas [30]. Measured concentrations of glyphosate in surface waters have ranged from 2.7 to 10 mg a.e./L [31]. Several studies have reported a wide range of toxic effects of glyphosate on aquatic organisms, especially freshwater organisms. In most of them, glyphosate can induce oxidative stress and DNA damage, inhibit enzymatic activities, and affect energy metabolism and growth rates [32,33]. Recent scientific data report the involvement of pesticides in the lipid composition modulation of aquatic organisms, although few studies have been conducted on bivalves [34,35,36,37,38]. Telahigue [38] investigated the toxic effect of Gly on scallops *Flexopecten glaber* in which they reported an altered FA profile.

In recent decades, the types of hazards to marine ecosystems have changed considerably, so in addition to emerging chemicals such as glyphosate, there are a lot of environmental stressors to which organisms are simultaneously exposed. Among them, rising water temperatures, associated with global climate change, represents an important concern for scientists and regulatory bodies worldwide. Environmental stressors interfere with cellular and molecular constituents; therefore, biochemical levels are sensitive indicators and respond quickly to stressors [39]. Several studies have shown that marine organisms actively change their fatty acid compositions, and thus their lipid quality, under environmental temperature changes [40,41,42]. In particular, the higher temperatures determine a reduction in unsaturated FA percentage to maintain the structural rigidity level of cell membranes in these thermal conditions [43,44]. 

Since there is a paucity of investigations related to the effects of multiple stressors on marine organisms that can be susceptible to combinational effects [45,46], it has become urgent to investigate the effects of emerging compounds, such as glyphosate, in the presence of environmental stressors on the nutritional quality of seafood. 

Alterations in fatty acid (FA) composition can be used as a sensitive early-warning bio-indicator of stress, providing important information regarding the combined effects of chemical and environmental perturbations on the nutritional qualities of a product.

Therefore, the aim of this study was to investigate the combined effects of increased culturing temperature (26 °C) and Gly on the lipid nutritional quality of the ecologically and economically important Mediterranean mussel *M. galloprovincialis*. Effects were evaluated based on lipid content and FA composition. Moreover, since this study aimed to provide useful information for consumers, and the determination of the FA profile alone is not sufficient to explain the lipid nutritional properties of a product, a number of lipid nutritional quality indices (LNQIs) were determined to be an important tool to assess the nutritional value and healthiness of foods. The LNQIs used included both traditional ones, such as the PUFA/SFA, UFA/SFA, and n-6/n-3 ratios as well as the atherogenic (AI), thrombogenic (TI), and hypocholesterolemic/hypercholesterolemic (h/H) indices, and more recent options, such as the health-promoting index (HPI), the unsaturation index (UI), the flesh lipid quality (FLQ), and the polyene index (PI). They allow us to explore more effectively and quickly the nutritional lipid properties of the mussel.

## 2. Materials and Methods

### 2.1. Chemicals and Reagents

The formulation of glyphosate (Gly [N-(phosphonomethyl)Glycine, C_3_H_8_NOP) used was Roundup^®^Platinum (Bayer CropScience S.r.l.), containing 480 g/L a.e. glyphosate (potassium salt, 43.8%) and other ingredients (56.2%). All reagents and solvents (methanol, chloroform, and n-hexane) used were of analytical HPLC and GC grade and were purchased from Sigma-Aldrich (Saint Louis, MO, USA). MilliQ water was used for all dilutions and the preparation of the reagent, and standard solution was obtained from Honeywell Riedel-de-Haën (Seelze, Germany). Extra-pure sodium sulfate anhydrous was obtained from Fluka (Sigma-Aldrich Chemie, Buchs, Switzerland). Helium 5.5 was obtained from Rivoira (Milan, Italy). Fatty Acids Methyl Esters (FAME) Mix standards 37 components and PUFA N° 1 were supplied by Supelco (Bellafonte, PA, USA), while methyl nonadecanoate (C19, internal standard, 98% purity) was purchased from Sigma–Aldrich^TM^.

### 2.2. Mussels and Exposure to GLY and Temperature (T°C)

*M. galloprovincialis* (4.5 ± 0.8 cm shell length; 5.8 ± 0.7 g wet weight) were collected from an unpolluted area [47] in the Second Inlet of the Mar Piccolo Sea in Taranto (Southern Italy), in January 2022. The mussels were transported in cold boxes with native seawater. At the laboratory, the mussels were randomly placed in large aquaria (20 L) and maintained with aerated seawater (salinity of 36 ± 1‰, temperature of 20 ± 0 °C) at a photoperiod of 12:12 h light:dark, without food, for an acclimatization period of 48 h, before experiencing exposure to Gly at concentrations of 1 and 10 mg/L.

Only mussels that showed apparent good health status (secretion of new byssal threads and reattachment to aquarium surface) were used for the exposure to Gly. A stock solution (0.1 g/L) of Gly was prepared in deionized water and stored in the dark at 4 °C. Work solutions were prepared immediately prior to mussel exposure by diluting the stock solution in 0.45 µm-filtered natural sea water (FNSW) (8.2 ± 0.1 pH, 8.0 ± 0.1 mg/L dissolved oxygen and 36‰ salinity). The mussels (30 per concentration) were exposed for 4 days to 0 mg/L (control, CTR), 1 mg/L, and 10 mg/L Gly and two temperatures (20 ± 1 and 26 ± 1 °C). The nominal concentrations were chosen based on data regarding Gly concentrations in aquatic ecosystems. The mussels (10 per tank) were kept in 10 L glass aquaria containing aerated seawater under a photoperiod of 12hL:12hD. Each bioassay was performed across three replicates per treatment. Bivalves were fed daily with a mix of microalgae *Isochrysis galbana*, *Tetraselmis suecica*, and *Chaetoceros calcitrans* (1 × 10^5^ cells/mL). Everyday seawater, Gly stock solution, and the Gly concentrations in the exposure tanks were renewed every 24 h. Moreover, the mussels were checked for mortality and valve condition, behavior during feeding, and byssus attachment. At the end of the experimental period, mussel soft tissues were separated from the shells and stored at −20 °C (for a maximum of 4 days).

### 2.3. Moisture, Lipid and Fatty Acid Analyses

Moisture was measured by oven drying at 105 °C to a constant weight. Total lipids (TL) were extracted according to Folch [48] and determined gravimetrically. Fatty acids were transesterified to methyl ester (FAMEs) according to the procedure described by Prato [49]. FAMEs were analyzed by gas chromatography (GC) using an HP 6890 series GC (Hewlett Packard, Wilmington, DE, USA) equipped with a flame ionization detector. FAMEs were separated with an Agilent HP-88 column (60 m × 0.25 mm id, film thickness 0.25 µm) by Agilent Technologies (Santa Clara, CA, USA). Helium was used as the carrier gas at a flow rate of 1 mL/min. FAMEs were identified by comparing retention times with reference standards (Supelco 37 Component FAME Mix, and PUFA N° 1). The relative quantities of FAs were calculated as percentages (% of total FAs). Quantification (mg/100 ww) was made using methyl nonadecanoate (50 µg/mL) as an internal standard. 

### 2.4. Lipid Nutritional Quality Indices (LNQIs)

The nutritional lipid quality of mussels (CTR and treated) was evaluated using nutritional indicators based on fatty acid composition. The PUFA/SFA, UNS/SFA, n-3/n-6 ratio, n-6/n-3 EPA + DHA, arachidonic acid (ARA, C20:4 n-6)/DHA ratio, and ARA/EPA indices were evaluated. In order to obtain a comprehensive evaluation of the lipid quality of mussels, the following indices were also calculated: the atherogenic index (AI), the trombogenic index (TI) [50], the hypocholesterolemic/hypercholesterolemic (h/H) ratio index [51], the health-promoting index (HPI), the unsaturation index (UI), the flesh lipid quality (FLQ) [52], and the polyene index (PI) [53]. These were determined by means of the following equations: $$AI=\frac{C12:0+4\times C14:0+C16:0}{\sum MUFAs+\sum PUFAs}$$
$$TI=\frac{\left(C14:0+C16:0+C18:0\right)}{\left[\left(0.5\times \sum MUFAs+0.5\times \sum n-6PUFAs+3\times \sum n-3PUFAs+\frac{n-3}{n-6}\right)\right]}$$
$$HH=\frac{\left(C18:1cis9+C18:2n-6+C20:4n-6+C18:3n-3+C20:5n-3+C22:5n-3+C22:6n-3\right)}{\left(C14:0+C16:0\right)}$$
$$HPI=\frac{\sum UFA}{\left[12:0+(14:0\times 4)+16:0\right]}$$
$$UI=1\times \left(\%monoenoics\right)+2\times \left(\%dienoics\right)+3\times \left(\%trienoics\right)+4\times \left(\%tetraenoics\right)+5\times \left(\%pentaenoics\right)+6\times \left(\%hexaenoics\right)$$
$$FLQ=100\times \frac{22:6n-3+20:5n-3}{\sum UFA}$$
$$PI=\frac{\left(20:5n-3+22:6n-3\right)}{16:0}$$

### 2.5. Statistical Analyses

The data, expressed as mean values ± standard deviation, were analyzed for normality and variance homogeneity through Kolmogorov–Smirnov and Levene’s tests, respectively. When either assumption was meet, a two-way ANOVA analysis was used to verify the effect of Gly, temperature, and their interaction on moisture, total lipid content, FA profile, and the LNQIs. Subsequently, post-hoc LSD multiple comparisons were performed. The level of significance was set as 0.05.

## 3. Results and Discussion

### 3.1. Effects of Temperature and Glyphosate on Moisture and Lipid Content of M. galloprovincialis

As shown in Figure 1, in general, CTR and treated mussels exhibited high moisture (%) and low lipid content (g/100 g w.w. basis), as demonstrated in previous studies [3,4,5,41]. Significant effects of T°C (F = 9.476, *p* < 0.05) and Gly (F = 9.909, *p* < 0.05) on moisture were observed, and in particular, mussels exposed to 10 mg/L Gly at 20 °C showed the highest value (89%).

Two-way ANOVAs performed on lipid content revealed the significant effects of T°C (F = 29.23, *p* < 0.05) and Gly (F = 25.25, *p* < 0.05), as well as significant interactions between T°C and Gly (F = 8.677, *p* < 0.05). However, CTRs did not show a significant difference in lipid content between the two temperatures, with values of 1.32 and 1.38 g/100 g ww, at 20 and 26 °C, respectively (*p* < 0.05). These results agree with previous studies [3,4,5,41]. Relative to the treatments at a temperature of 20 °C, the mussels exposed to Gly 1 and 10 mg/L showed significant decreases in TL (*p* < 0.05), with values of 1.16 and 1.15 g/100 g ww, respectively. At a temperature of 26 °C, mussels exposed to 10 mg/L Gly showed the same trend with a significant decrease in TL (1.19 g/100 g ww), while mussels exposed to 1 mg/L Gly did not differ significantly from the CTR (*p* > 0.05). Frontera [54] and Avigliano [55] reported that high glyphosate concentrations are able to reduce lipid reserves in many crustacean species.

### 3.2. Effects of Temperature and Glyphosate on Fatty Acids of M. galloprovincialis

The fatty acids, expressed as % of total FAs and mg/100 g ww, of CTR and of mussels exposed to Gly at two temperatures are reported in Table 1 and Table 2. Twenty fatty acids exceeding a minimum of 0.1% of total FAs were identified. Despite mussels having low lipid contents, they showed FA profiles beneficial to human health in all samples. 

However, the mussels exhibited changes in FA composition after exposure to Gly (1 and 10 mg/L) at the two different temperatures (20 °C and 26 °C) (two-way ANOVA, *p* < 0.05).

The results showed a dominance of SFA (range 41–49%) and PUFA (range 30–38%) in all samples, followed by lower amounts of MUFA (range 18–24%), though with significant differences among treatments (Figure 2, Table 1). The FA contents of the CTRs were similar to those reported in previous studies on *M. galloprovincialis* by Biandolino [3,4] and Prato [5]. 

Two-way ANOVA showed significant interaction effects of T°C and Gly on SFA (F = 4.263, *p* < 0.05) and PUFA (F = 5.449, *p* < 0.05) contents, whereas T°C (F = 46.44, *p* < 0.05) and Gly (32.07, *p* < 0.05) individually had significant effects on MUFA content, on which their interaction had no effect (F = 0.48, *p* > 0.05). Even though temperature did not have a significant effect on SFA level, the CTR mussels at 26° C had a slightly higher SFA content. This is in accordance with results reported by Fadhlaoui and Lavoie [36] in a similar study on the effects of temperature and Gly on FA composition of the gastropod *Lymneae* sp. One explanation could be an adaptive response to maintain the structural rigidity of the cell membrane in these conditions [34,37]. Mussels exposed to the most Gly (10 mg/L) at 20 °C showed the highest level of SFA (49% of total FAs, Table 1), due to the increased proportions of stearic (C18:0) and palmitic (C16:0) acids. Similarly, Telahigue [38] reported an enhanced saturation level in the digestive gland of *F. glaber* exposed to Gly at 18 ± 1 °C. 

An increase in temperature induced a significant increase in MUFA (Figure 2, Table 1) as evidenced by the higher levels in the CTR at 26 °C compared with those at 20° C. This was probably due to a defense mechanism to overcome oxidative cell damage after exposure to thermal stress [56].

The content of the major MUFA, oleic acid (C18:1 n-9), increased from 1.18 to 4.57% of total FAs at 20 and 26 °C, respectively (Table 1). Previous studies have demonstrated that aquatic invertebrates change their fatty acid profile, thus altering their lipid quality, in response to various environmental stressors [57,58]. The importance of temperature as a controlling factor for aquatic organisms is well known. It can affect the physiology of organisms and their metabolic rates, as well as the toxicity of contaminants. In a similar study, Fadhlaoui and Lavoie [36] reported a different trend showing a decrease in oleic acid at the highest temperature. However, Cohen [59], in a study on the effects of the herbicides on FAs, reported an increase in oleic acid in one cyanobacterium and one marine microalga, thus indicating an inhibitory effect of the herbicide on the Δ6-desaturation system. 

In the same way, exposure to Gly caused a significant increase in MUFA compared with CTRs in this study (*p* < 0.05, Figure 2, Table 1 and Table 2). Similarly, enhanced levels of MUFA have been found in the gills of *F. glaber* exposed to Gly at 18 ± 1 °C [38]. 

With regard to PUFA, a significant difference was observed between CTRs according to the temperature, with a significant decrease at 26 °C (*p* < 0.05, Table 2), indicating that mussels use FAs as source of energy to defend against thermal stress, which was also indicated by Anacleto [24] for two clam species. Several studies on marine organisms have reported an inverse relationship between the PUFA content and temperature since this class of FA plays a crucial role in maintaining the fluidity of the cell membrane in mollusks when the temperature is low [60,61,62].

Moreover, a significant reduction in PUFA level in mussel exposed to 10 mg/L Gly at 20 °C was also observed (*p* < 0.05). Similarly, Telhaigue [38] reported a decrease in PUFA in the digestive gland of *F. glaber* exposed to Gly at 18 ± 1 °C. Gly did not appear to have any significant effect on PUFA in mussel kept at 26 °C (*p* > 0.05). 

In general, in all samples, palmitic acid (C16:0) was the most abundant SFA (range 26–32% of total FAs) followed by myristic (C14:0) and stearic (C18:0) acid (Table 1 and Table 2). Palmitoleic (C16:1) and oleic (C18:1 n-9) acid were the predominant MUFAs, whereas, among PUFAs, n-3 EPA and DHA were the most abundant, accounting for about 73–88% of total PUFAs.

Data from the literature report the FA profiles of various mollusks bivalves, such as mussels, clams, scallops, and oysters [3,5,6,63], but information on their variation under combined chemical and thermal stress are lacking. In response to Gly and/or temperature exposure, changes in individual FAs were also observed (*p* < 0.05), such as the n-6 PUFA ARA, which showed a significant increase at the highest concentration at 26 °C (*p* < 0.05, Table 1). This increase may reflect a selective retention of ARA due to its role in the cell transmission of signals associated with tissue inflammation and adaptation to environmental stress [64].

Two-way ANOVA revealed the significant effect of T°C × Gly interaction on the most nutritionally important fatty acids, such as EPA and DHA. These n-3 PUFAs are involved in fundamental physiological processes and have many positive effects on human health [65].

Increased water temperature greatly influences the physiology and metabolic processes of marine organisms [66] and may thus modify lipid bioconversion capacity, impacting n-3 PUFA content and consequently the organisms’ nutritional value. On the other hand, FA profiles showed modifications in mussels exposed to Gly, probably due to the interference of Gly during the biosynthesis of lipids, as suggested by Telhaigue [37]. Additionally, Cohen [59] demonstrated that pesticides alter FA biosynthesis, acting as potent inhibitors of the Δ6-desaturation system and the biosynthesis of long-chain FAs.

CTRs showed a significant decrease in EPA at 26 °C compared with CTRs at 20 °C (*p* < 0.05, Table 1). Moreover, the mussels showed a significant decrease in EPA content after exposure to the highest Gly concentration (10 mg/L) at both temperatures (*p* < 0.05). According to Cohen [59], who observed a similar inhibition of EPA, this apparently occurs due to a blockage of the desaturation of C18:2 to C18:3 n-3 in one pathway of EPA biosynthesis. Similarly, the level of DHA decreased due to the effect of the T°C × Gly interaction (F = 4.182, *p* = 0.041). Bayir [67], in a study on brown trout (*Salmo trutta*), reported a significant decrease in EPA and DHA after exposure to Gly (10 and 20 mg/L). Telhaigue [38] showed a significant decrease in EPA and DHA in the digestive gland of *F. glaber* after glyphosate exposure at 18 ± 1 °C, while an opposite trend was found in the gills. On the other hand, Fadhlaoui and Lavoie [36] found no significant effect of Gly and temperature on EPA in snails. 

### 3.3. Lipid Nutritional Quality Indices

Since the different FAs have distinctive functional roles, in this study, a number of lipid nutritional quality indices were applied, with the purpose of assessing the healthiness of the lipid fraction of CTR and treated mussels. The results, reported in Table 3, revealed that exposure to Gly and thermal stress can significantly reduce the lipid nutritional value of *M. galloprovincialis* (*p* < 0.05). 

As reported above, EPA and DHA play a fundamental role in the prevention and treatment of many diseases, and since the quantity of EPA and DHA is higher in seafood, the sum EPA + DHA is often used to assess the nutritional quality of marine animal products. In the present study, EPA + DHA varied in the range of 18 to 25% of total FAs (*p* < 0.05), with the lowest value present in mussels exposed to 10 mg/L Gly at 20 °C. Despite this, the EPA + DHA values were comparable to values reported by Prato [6] for *M. galloprovincialis* and other bivalve species.

Both n-3- and n-6-series PUFAs are precursors to eicosanoids, which regulate inflammation in opposite ways [68]. n-6 derivatives promote inflammation, platelet aggregation, and vasoconstriction (pro-inflammatory reaction), while n-3 derivatives inhibit inflammation and platelet aggregation and enhance vasodilation (anti-inflammatory and resolving reaction) [68]. Therefore, it is crucial to maintain a balanced ratio of these two PUFA families in our diet for optimal human growth, human wellbeing, and the prevention of the onset of modern chronic diseases. However, more n-6 than n-3 is consumed in the modern westernized diet, shifting the balance in favor of n-6, producing a consequent n-6/n-3 ratio of 17:1 or 20:1 instead of 1:1, and resulting in a negative effect on human health [69].

The n-6/n-3 ratio and its inverse are among the most used indices as valuable indicators of the lipid quality of foods and diets. To be considered optimal, the recommended n-6/n-3 ratio in a diet should range from 1/1 to 4/1 [70]. Lower values indicate a very healthy and useful food in the prevention of coronary heart disease.

In this study, n-6/n-3 ratios ranged from 0.18 to 0.22 and showed significant differences (*p* < 0.05) between CTR and Gly-exposed groups at 26 °C. Despite this, the values can be considered favorable (<1) independent from treatments, confirming the fact that mussels are a seafood characterized by an optimal n-6/n-3 ratio compared to most terrestrial-origin food [70]. Gly exposure had significant effects on the n-3/n-6 ratio (F = 7.832, *p* = 0.006), which markedly decreased compared to CTR mussels at both temperatures; however, all samples, CTRs and exposed samples, exhibited values in the range of 5.45–4.48, which is better than recommended.

The PUFA/SFA ratio is a widely used index to evaluate the impact of diet on cardiovascular health and oxidative stress [52]. In this study, a significant decrease in this ratio in mussels exposed to 10 mg/L Gly at 20 °C (0.61) compared to CTR mussels at 20 °C (0.85) was observed, even though all samples exhibited values above the recommended minimum value of 0.45 and are therefore favorable for human health [70]. However, some researchers consider PUFA/SFA to be weak and inadequate as an indicator of the nutritional value of fats for different reasons. The sum of PUFA does not distinguish between n-3 and n-6; therefore, a high proportion of PUFA is not necessarily healthy if it is not balanced in relation to the PUFA n-6/n-3 ratio. Moreover, the favorable influence of MUFA (i.e., increasing lipoprotein receptor activity and decreasing serum cholesterol concentration) is not considered by this index. In addition, among SFA, stearic acid (C18:0) does not increase the concentration of plasma cholesterol and low-density lipoproteins (LDLs) [70].

The UNS/SFA ratio also takes into account the health benefits of MUFAs for preventing cardiovascular risk. The highest UNS/SFA ratio was observed in mussels exposed to Gly at 26 °C, mostly due to the low SFA and high MUFA contents at this temperature. Anacleto [24] found similar results for two *Ruditapes* species exposed to thermal stress. Snails (*Lymneae* sp.) exposed to Gly and thermal stress did not show significant differences among treatments [36] in the UNS/SFA ratio. 

n-3 EPA and DHA compete with n-6 ARA for the synthesis of eicosanoids. Therefore, concentrations of EPA and DHA higher than ARA shift the process balance toward less inflammatory activity. For this reason, the ratios ARA/DHA and ARA/EPA are both important, simple, rapid, and reliable indices for determining n-3 fatty acid status. In this study, exposure to Gly caused a significant increase in the ARA/DHA ratio (*p* < 0.05), while the ARA/EPA ratio exhibited a significant increase when exposed to both the highest temperature and Gly concentration (*p* < 0.05). This is probably due to an adaptive response to the inflammation process since ARA plays an essential role in the immunological system.

Another approach to evaluating the dietary quality of the lipid fraction for human health is the use of the AI and TI indices, considered to be reliable indicators of lipid quality of food, more specific compared to PUFA/SFA. AI has evidenced the relationship between SFA and UNS and the atherogenic potential of the FA content [50]. For the determination of AI, the authors did not include stearic acid (C18:0) among SFAs since it is not considered pro-atherogenic due to its rapid conversion to oleic acid in the human body. Based on the relationship between the prothrombogenic and anti-prothrombogenic FAs, TI shows tendency to form clots in the blood vessels. Lower values of these indices indicate the presence of FAs with anti-atherogenic and anti-thrombogenic proprieties; therefore, they have a higher potential to prevent coronary heart disease [50]. AI and TI values less than 1.0 and 0.5, respectively, are recommended in the diet [71]. In this study, AI and TI exhibited values in the ranges of 0.83–1.18 and 0.37–0.50, respectively, and were thus within the ranges of values reported in previous studies for the same species and other bivalves [3,4]. These values are beneficial to human health. These indices did not significantly vary with temperature; a statistically significant difference in AI was found only between CTR mussels and those exposed to Gly at 26 °C (*p* < 0.05). TI showed a significant highest value in mussels exposed to 10 mg/L of Gly at 20 °C.

A significant effect of Gly (F = 6.993, *p* = 0.01) and Gly × T°C interaction (F = 4.522, *p* = 0.03) on the h/H index was found. The h/H ratio assesses the effect of specific dietary FAs on cholesterol metabolism, and high values indicate healthier food, desirable for human health benefit. Mussels exposed to 10 mg/L of Gly at 20 °C and CTR mussels at 26 °C (Table 3) exhibited the lowest values. The results from this study are consistent with previously published data on different seafood species [62].

The health-promoting index (HPI) is another index that estimates the nutritional value of dietary fat and can be used to assess the potential effects of FAs on cardiovascular health. The higher this index, the more positive the effect in preventing cardiovascular diseases [52]. The results obtained for CTR and treated *M. galloprovincialis* showed HPI values between 0.86 and 1.24 (*p* < 0.05, Table 3), much higher than the values reported by Chen and Liu [52] for dairy products (range 0.16–0.68); therefore, eating mussel may be more beneficial for human health than dairy products. On the other hand, a 2021 study by Araujo et al. [72] on the fish *Trematomus bernacchii* from the Antarctic region found HPI values ranging from 1.88 to 2.12.

The UI is used to evaluate the content of high-quality PUFA. In this index, a different weight is attributed to each of the different unsaturated fatty acids, so a high value means a high degree of total unsaturation, which is favorable for maintaining the lipid membrane fluidity. However, the weakness of this index is that it does not distinguish the two different fatty acids series, n-3 and n-6. In this study, a significant difference was observed for the CTRs according to the temperature, showing a decrease at 26 °C (*p* < 0.05, Table 3) due to the need to maintain the structural rigidity of the cell membrane under thermal stress. Moreover, mussels exposed to 10 mg Gly/L at 20 °C showed the lowest value (150) (*p* < 0.05). The UI is commonly used for macroalgae, but its use in meat, crops, and dairy products has also been reported. In particular, UI values range from 45 to 369 in macroalgae, from 125 to 155 in crops, from 73 to 124 in meat, and from 86 to 126 in dairy products [52]. The UI values determined in this study for CTR and treated mussels were generally higher than those found in other products.

Flesh lipid quality (FLQ) is an index more suitable for marine products since it refers to the percentage correlation between the sum EPA + DHA and the total FAs. A higher value indicates a higher quality of dietary lipid source. The polyene index (PI) is an index normally used to measure PUFA damage [53]. As shown in Table 3, both indices, FLQ and PI, followed the same trend as UI, showing the lowest values at the higher Gly concentration at 20 °C and in CTR mussels at 26 °C (*p* < 0.05). 

The FLQ values are within the range reported in the literature for freshwater and marine fish (range 13–33) [73]. Łuczýnska [73] found the following FLQ ratio values: 30.14 in bream, 33.22 in perch, 24.32 in ide, 16.99 in carp, 17.97 in rainbow trout, 20.25 in flounder, and 13.01 in herring.

Additionally, PI values in the CTR and treated mussels exhibited better values than those reported in the literature for golden grey mullet (0.32–0.55) and gold band goatfish (0.50–0.56) [74].

#### EPA + DHA

The EPA + DHA sum (mg/100 g ww) is one of the most important lipid nutritional quality indices, providing important information on their absolute quantity. Given the recognized potential health benefits of EPA and DHA in human metabolism, government and medical agencies worldwide recommend a regular seafood consumption of about 500 mg/day of EPA + DHA to reduce the risk of cardiovascular disease [75,76,77,78]. 

Previous studies investigating the effects of glyphosate exposure on different shellfish species have focused on endpoints related to changes in antioxidant capacity and general metabolic enzymes [29]. Studies evaluating the effects of pesticides on nutritional lipid quality are lacking.

Mussels have been suggested as an excellent dietary source of EPA and DHA for human consumers [1] due to their essential function.

Two-way ANOVA performed on the EPA + DHA quantity revealed no significant effect of temperature (F = 0.657, *p* > 0.05), although at 26 °C the value was slightly lower than that at 20 °C.

Gly (F = 5.143, *p* < 0.05) and T × Gly interaction (F = 6.859, *p* < 0.05) significantly affected EPA + DHA. The lowest level was found in mussels exposed to 10 mg/L Gly at 20 °C with a value of 150 mg/100 g ww (Table 4); therefore, a serving of 167–334 g is needed to satisfy the daily intake of EPA + DHA as recommended by the WHO (250–500 mg) [78]. Meanwhile, a serving of 110–220 g of CTR mussels’ group at 20 °C, containing 228 mg/100 g of EPA + DHA, is enough to cover the EPA + DHA requirement for a health benefit. Obviously, mussels cannot represent the only source of n-3 in a balanced diet; however, this information is interesting, considering the low-fat content of mussels. 

## 4. Conclusions

The results obtained in this study clearly demonstrate that a short-term exposure to both stressors, glyphosate and a rise in temperature, could change the lipid content and FA composition of the mussel *Mytilus galloprovincialis*.

The mussels’ lipid nutritional quality decreased, as can be seen in the applied LNQIs, which were shown to play a key role in understanding the human health benefits of consuming mussels.

The changes observed in the n-3/n-6 ratios, EPA + DHA, PUFA/SFA, AI, TI, PI etc., which mostly occurred in the groups exposed to 10 mg/L of glyphosate at a temperature of 20 °C and in those exposed to a temperature of 26 °C, indicated the loss of the mussels’ nutritional quality. These findings provide further evidence of the potential toxic effect of glyphosate on marine organisms, pointing to potential harmful effects to consumers resulting from the use of organophosphorus herbicides.

This study highlights the importance of undertaking further research including chronic exposures that will allow us to predict the impacts on communities and thus on aquatic ecosystems and food quality.

## Figures and Tables

**Figure 1 foods-12-01595-f001:**
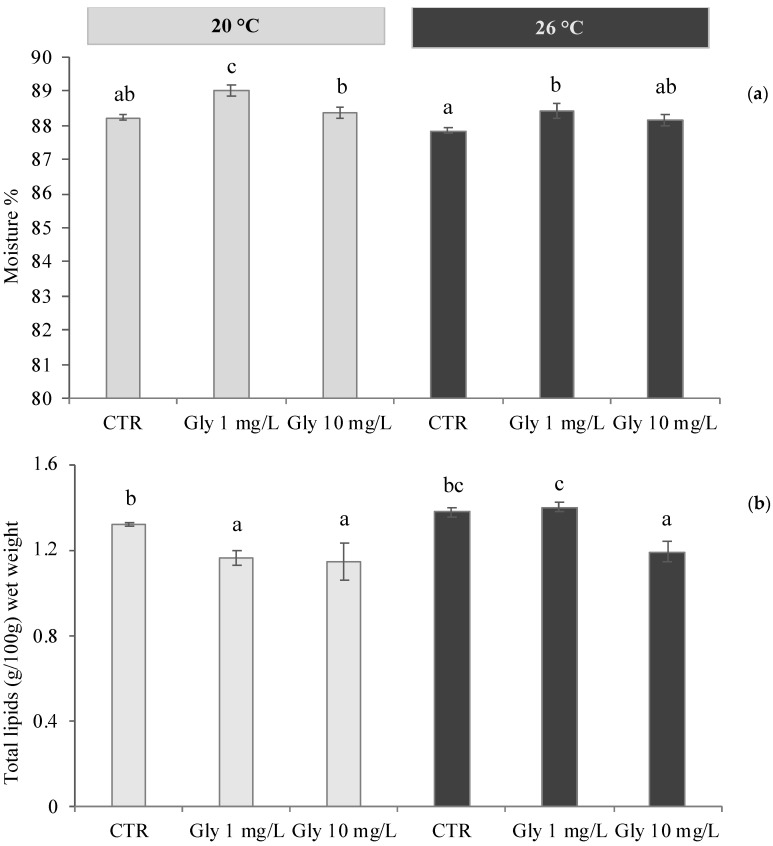
(**a**) Moisture (%) and (**b**) total lipids (g/100 g w.w. basis) of mussel among different exposure conditions in the control group (CTR) and glyphosate-exposed group (Gly) at 20 °C and 26 °C. Values are shown as mean ± s.d. Different letters over bars indicate significant differences among means (*p* < 0.05).

**Figure 2 foods-12-01595-f002:**
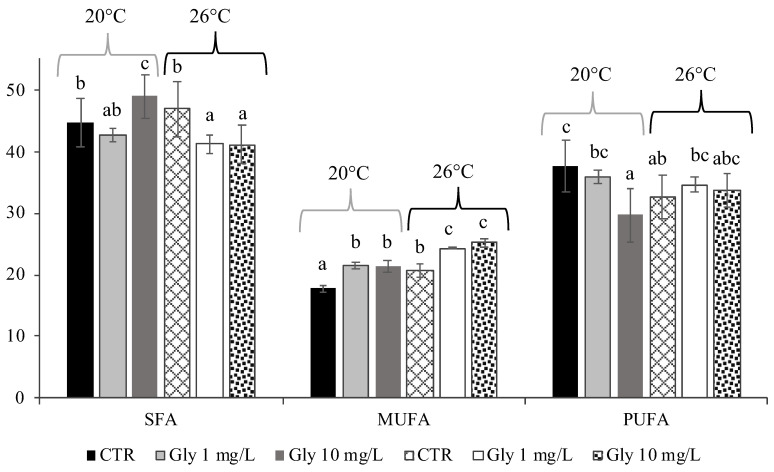
Proportion of saturated (SFA), monounsaturated (MUFA), and polyunsaturated fatty acids (PUFA) in mussels exposed to 1 and 10 mg/L Gly at 20 and 26 °C. Means with different letters on the bars differ significantly (*p* < 0.05).

**Table 1 foods-12-01595-t001:** Fatty acid profiles (% of total FAs) of mussels exposed to 1 and 10 mg/L Gly at 20 and 26 °C.

	20 °C			26 °C		
	CTR	Gly 1 mg/L	Gly 10 mg/L	CTR	Gly 1 mg/L	Gly 10 mg/L
C14:0	6.52 ± 1.28	5.59 ± 0.58	6.86 ± 0.30	7.74 ± 1.32	6.85 ± 0.27	5.52 ± 0.78
C15:0	1.59 ± 0.18	1.18 ± 0.17	1.43 ± 0.12	1.20 ± 0.18	1.21 ± 0.12	1.32 ± 0.48
C16:0	28.5 ± 2.9	27.5 ± 2.4	31.8 ± 0.6	30.9 ± 3.2	27.1 ± 1.4	26.1 ± 3.3
C17:0	1.72 ± 0.25 ^ab^	1.96 ± 0.31 ^bc^	2.17 ± 0.13 ^c^	1.61 ± 0.03 ^ab^	1.52 ± 0.08 ^a^	1.92 ± 0.24 ^bc^
C18:0	6.30 ± 0.35 ^c^	6.40 ± 0.23 ^c^	6.68 ± 0.218 ^c^	5.43 ± 0.29 ^b^	4.48 ± 0.21 ^a^	6.29 ± 0.26 ^c^
∑SFA	44.6 ± 2.9 ^b^	42.6 ± 3.5 ^ab^	48.9 ± 1.0 ^c^	46.9 ± 4 ^b^	41.2 ± 1.5 ^a^	41.1 ± 4.1 ^a^
C14:1	2.25 ± 0.19 ^b^	2.57 ± 0.58 ^b^	4.36 ± 0.34 ^c^	0.91 ± 0.12 ^a^	2.09 ± 0.32 ^b^	2.27 ± 0.22 ^b^
C15:1	2.36 ± 0.38 ^b^	2.67 ± 0.57 ^b^	3.54 ± 0.46 ^c^	1.48 ± 0.32 ^a^	2.10 ± 0.11 ^ab^	1.94 ± 0.37 ^ab^
C16:1	6.30 ± 0.80 ^a^	5.64 ± 0.15 ^a^	6.12 ± 0.38 ^a^	8.09 ± 0.61 ^b^	8.73 ± 0.326 ^b^	6.13 ± 0.98 ^a^
C17:1	0.59 ± 0.12 ^ab^	0.61 ± 0.17 ^ab^	0.58 ± 0.10 ^ab^	0.81 ± 0.07 ^bc^	0.86 ± 0.11 ^c^	0.49 ± 0.03 ^a^
C18:1 n-7	3.17 ± 0.06 ^ab^	3.41 ± 0.23 ^c^	3.40 ± 0.16 ^c^	3.04 ± 0.01 ^a^	2.98 ± 0.09 ^a^	3.29 ± 0.05 ^bc^
C18:1 n-9	1.18 ± 0.36 ^a^	4.44 ± 0.88 ^b^	1.91 ± 0.35 ^a^	4.57 ± 1.41 ^b^	5.34 ± 0.91 ^b^	8.80 ± 1.29 ^c^
C20:1 n-9	1.98 ± 0.25 ^bc^	2.11 ± 0.17 ^c^	1.52 ± 0.03 ^a^	1.70 ± 0.21 ^ab^	2.18 ± 0.13 ^c^	2.31 ± 0.25 ^c^
∑MUFA	17.8 ± 0.5 ^a^	21.5 ± 0.9 ^b^	21.4 ± 0.51 ^b^	20.6 ± 1.1 ^b^	24.3 ± 0.7 ^c^	25.2 ± 1.6 ^c^
C18:2 n-6	2.28 ± 0.06 ^ab^	2.50 ± 0.13 ^c^	2.18 ± 0.10 ^a^	2.13 ± 0.09 ^a^	2.37 ± 0.07 ^bc^	2.12 ± 0.08 ^a^
C18:3 n-3	4.10 ± 0.37 ^ab^	3.85 ± 0.32 ^a^	3.69 ± 0.46 ^a^	3.98 ± 0.08 ^a^	4.58 ± 0.25 ^b^	3.79 ± 0.11 ^a^
C18:4 n-3	2.05 ± 0.31	2.19 ± 0.10	1.74 ± 0.08	2.16 ± 0.08	1.77 ± 0.15	2.14 ± 0.45
C22:0 + 20:3 n-6	0.54 ± 0.10 ^bc^	0.62 ± 0.09 ^c^	0.40 ± 0.02 ^ab^	0.32 ± 0.02 ^a^	0.66 ± 0.13 ^c^	0.56 ± 0.11 ^bc^
C20:4 n-6 (ARA)	3.24 ± 0.18 ^bc^	3.41 ± 0.40 ^c^	2.76 ± 0.16 ^ab^	2.58 ± 0.27 ^a^	3.01 ± 0.06 ^bc^	3.27 ± 0.53 ^bc^
C20:5 n-3 (EPA)	14.6 ± 1.1 ^c^	14 ± 1.4 ^b^	12.0 ± 0.7 ^a^	13 ± 1.4 ^b^	13.4 ± 0.4 ^b^	11.7 ± 0.9 ^a^
C22:5 n-3	0.68 ± 0.11 ^a^	0.62 ± 0.14 ^a^	0.50 ± 0.03 ^a^	0.60 ± 0.10 ^a^	0.66 ± 0.07 ^a^	0.93 ± 0.12 ^b^
C22:6n3 (DHA)	10.0 ± 1.8 ^c^	8.73 ± 1.45 ^bc^	6.36 ± 0.85 ^a^	7.77 ± 0.87 ^ab^	8.10 ± 0.77 ^ab^	9.13 ± 1.38 ^bc^
∑ PUFA	37.5 ± 3.1 ^c^	35.9 ± 3.4 ^bc^	29.7 ± 1.0 ^a^	32.5 ± 2.4 ^ab^	34.5 ± 1.5 ^bc^	33.6 ± 3.0 ^abc^
n-3	31.4 ± 2.8 ^b^	29.4 ± 3.1 ^b^	24.3 ± 1.2 ^a^	27.5 ± 2.3 ^ab^	28.5 ± 1.4 ^b^	27.7 ± 2.3 ^ab^
n-6	6.07 ± 0.33 ^c^	6.53 ± 0.30 ^c^	5.34 ± 0.17 ^ab^	5.03 ± 0.20 ^a^	6.04 ± 0.17 ^c^	5.95 ± 0.61 ^bc^

All values are means of three separate replicates. Means with different letters (a, b, c) within each row indicate statistical differences (*p* < 0.05). SFA, saturated fatty acids; MUFA, monounsaturated fatty acids; PUFA, polyunsaturated fatty acids; ARA, arachidonic acid; EPA, eicosapentaenoic acid; DHA, docosahexaenoic acid.

**Table 2 foods-12-01595-t002:** Fatty acid profiles (mg/100 g ww) of mussels exposed to 1 and 10 mg/L Gly at 20 and 26 °C.

	20 °C			26 °C		
	CTR	Gly 1 mg/L	Gly 10 mg/L	CTR	Gly 1 mg/L	Gly 10 mg/L
C14:0	60.2 ± 11.2 ^bc^	44.8 ± 4.6 ^a^	55.9 ± 2.5 ^abc^	74.6 ± 9.5 ^d^	67.1 ± 3.1 ^cd^	46.1 ± 3.4 ^ab^
C15:0	14.7 ± 1.6	9.50 ± 1.12	11.7 ± 1.0	11.6 ± 1.4	11.9 ± 0.9	11.0 ± 1.2
C16:0	263 ± 27 ^c^	220 ± 19.5 ^ab^	259 ± 4.6 ^bc^	297 ± 14.8 ^c^	266 ± 11.6 ^c^	218 ± 12.7 ^a^
C17:0	15.9 ± 2.3	15.7 ± 1.5	17.7 ± 1.1	15.5 ± 0.9	14.9 ± 0.6	16.0 ± 1.1
C18:0	58.2 ± 3.2 ^c^	51.3 ± 1.8 ^b^	54.4 ± 1.4 ^bc^	53.3 ± 2.1 ^b^	44 ± 1.8 ^a^	52.4 ± 1.7 ^b^
∑SFA	412 ± 36 ^b^	342 ± 28 ^a^	398 ± 8.3 ^b^	451 ± 16.8 ^b^	403 ± 18.7 ^b^	343 ± 17.8 ^a^
C14:1	20.8 ± 1.7 ^b^	20.6 ± 3.1 ^b^	35.5 ± 2.7 ^c^	8.79 ± 0.84 ^a^	20.5 ± 1.8 ^b^	19.0 ± 1.8 ^b^
C15:1	21.8 ± 3.5 ^b^	21.4 ± 2.5 ^b^	28.9 ± 3.7 ^c^	14.3 ± 0.7 ^a^	20.6 ± 1.5 ^b^	16.2 ± 1.9 ^ab^
C16:1	58.2 ± 7.3 ^b^	45.2 ± 1.2 ^a^	49.9 ± 3.1 ^ab^	78 ± 4.9 ^c^	85.6 ± 1.9 ^c^	51.1 ± 3.1 ^ab^
C17:1	5.44 ± 1.12 ^a^	4.89 ± 1.07 ^a^	4.75 ± 0.83 ^a^	7.83 ± 0.53 ^b^	8.42 ± 0.76 ^b^	4.11 ± 0.36 ^a^
C18:1n7	29.3 ± 0.6	27.3 ± 1.8	27.7 ± 1.3	29.3 ± 0.4	29.2 ± 1.3	27.4 ± 1.2
C18:1n9c	10.9 ± 2.2 ^a^	35.6 ± 5.7 ^bc^	15.6 ± 2.1 ^ab^	44 ± 6.8 ^c^	52.3 ± 2.8 ^c^	73.3 ± 8.3 ^d^
C20:1n9	18.3 ± 2.2 ^bc^	16.9 ± 1.4 ^b^	12.4 ± 0.3 ^a^	16.4 ± 1.5 ^b^	21.3 ± 1.5 ^c^	19.2 ± 1.4 ^bc^
∑MUFA	164 ± 4.9 ^a^	172 ± 7.2 ^a^	174 ± 4.1 ^a^	199 ± 8.9 ^b^	238 ± 7.2 ^c^	210 ± 11.7 ^b^
C18:2n6c	21.1 ± 0.6 ^b^	20.0 ± 1.1 ^b^	17.8 ± 0.8 ^a^	20.5 ± 0.5 ^b^	23.2 ± 0.6 ^c^	17.7 ± 0.9 ^a^
C18:3n3	37.8 ± 3.5 ^b^	30.9 ± 2.5 ^a^	30.1 ± 3.2 ^a^	38.3 ± 1.3 ^b^	44.9 ± 3.1 ^c^	31.6 ± 1.3 ^a^
C18:4n3	19.9 ± 3.7	17.5 ± 0.8	14.2 ± 0.6	20.9 ± 1.1	17.3 ± 1.5	17.9 ± 1.2
C22:0 + 20:3n6	5.04 ± 0.98 ^cd^	5.00 ± 1.01 ^cd^	3.2 ± 0.2 ^ab^	3.11 ± 0.31 ^a^	6.53 ± 0.58 ^d^	4.71 ± 0.83 ^bc^
C20:4n6	30 ± 2.7 ^c^	27.3 ± 3.2 ^abc^	22.5 ± 1.3 ^a^	24.9 ± 1.6 ^ab^	29.5 ± 1.2 ^bc^	27.3 ± 2.8 ^abc^
C20:5n3 (EPA)	135 ± 11.7 ^c^	112 ± 11.3 ^ab^	98.0 ± 6.0 ^a^	125 ± 9.6 ^bc^	131 ± 3.8 ^bc^	97.3 ± 8.4 ^a^
C22:5n3	6.32 ± 1.67 ^bc^	4.94 ± 0.94 ^ab^	4.07 ± 0.29 ^a^	5.78 ± 0.74 ^abc^	6.46 ± 0.51 ^bc^	7.76 ± 0.62 ^c^
C22:6n3 (DHA)	92.7 ± 11.3 ^c^	70.0 ± 10.1 ^b^	51.8 ± 6.1 ^a^	74.9 ± 8.3 ^bc^	79.4 ± 5.4 ^bc^	76.2 ± 9.6 ^bc^
∑ PUFA	346 ± 31.7 ^d^	288 ± 24.7 ^abc^	242 ± 8.2 ^d^	314 ± 11.9 ^bcd^	339 ± 12.6 ^cd^	280 ± 11.3 ^ab^
n-3	291 ± 21.3 ^d^	235 ± 14.6 ^ab^	198 ± 9.5 ^a^	265 ± 18.6 ^cd^	279 ± 11.7 ^cd^	231 ± 21.1 ^ab^
n-6	56.1 ± 3.1 ^cd^	52.4 ± 2.3 ^bc^	43.5 ± 1.3 ^a^	48.5 ± 2.1 ^ab^	59.2 ± 2.3 ^d^	49.7 ± 3.7 ^b^

All values are means of three separate replicates. Means with different letters (a, b, c, d) within each row indicate statistical differences. (*p* < 0.05). SFA, saturated fatty acids; MUFA, monounsaturated fatty acids; PUFA, polyunsaturated fatty acids; ARA, arachidonic acid; EPA, eicosapentaenoic acid; DHA, docosahexaenoic acid.

**Table 3 foods-12-01595-t003:** Lipid nutritional quality indices of *M. galloprovincialis* in control group (CTR) and glyphosate-exposed group (Gly) at 20 and 26 °C. Values are means ± standard deviations.

	20 °C			26 °C		
	CTR	Gly 1 mg/L	Gly 10 mg/L	CTR	Gly 1 mg/L	Gly 10 mg/L
EPA + DHA %	24.6 ± 2.4 ^c^	22.7 ± 2.3 ^bc^	18.4 ± 1.5 ^a^	20.8 ± 2.3 ^ab^	21.5 ± 1.1 ^abc^	20.8 ± 2.0 ^ab^
n-3/n-6	5.17 ± 0.47 ^bc^	4.48 ± 0.32 ^a^	4.56 ± 0.25 ^a^	5.45 ± 0.45 ^c^	4.71 ± 0.14 ^ab^	4.63 ± 0.13 ^a^
n-6/n-3	0.19 ± 0.02 ^ab^	0.22 ± 0.02 ^bc^	0.22 ± 0.02 ^bc^	0.18 ± 0.02 ^a^	0.21 ± 0.01 ^bc^	0.21 ± 0.01 ^bc^
PUFA/SFA	0.85 ± 0.12 ^b^	0.85 ± 0.10 ^b^	0.61 ± 0.03 ^a^	0.70 ± 0.11 ^ab^	0.84 ± 0.10 ^b^	0.83 ± 0.09 ^b^
UNS/SFA	1.25 ± 0.16 ^abc^	1.35 ± 0.11 ^bc^	1.04 ± 0.04 ^a^	1.15 ± 0.16 ^ab^	1.43 ± 0.09 ^c^	1.44 ± 0.15 ^c^
ARA/DHA	0.33 ± 0.03 ^a^	0.40 ± 0.06 ^b^	0.44 ± 0.08 ^b^	0.34 ± 0.05 ^a^	0.37 ± 0.03 ^b^	0.36 ± 0.02 ^ab^
ARA/EPA	0.22 ± 0.02 ^ab^	0.24 ± 0.00 ^b^	0.23 ± 0.03 ^b^	0.20 ± 0.00 ^a^	0.22 ± 0.01 ^ab^	0.28 ± 0.01 ^c^
AI	0.99 ± 0.11 ^ab^	0.87 ± 0.13 ^a^	1.16 ± 0.06 ^b^	1.18 ± 0.15 ^b^	0.93 ± 0.07 ^a^	0.83 ± 0.16 ^a^
TI	0.38 ± 0.06 ^a^	0.38 ± 0.05 ^a^	0.50 ± 0.03 ^b^	0.44 ± 0.04 ^ab^	0.37 ± 0.03 ^a^	0.37 ± 0.08 ^a^
h/H	1.05 ± 0.15 ^bc^	1.15 ± 0.14 ^bc^	0.76 ± 0.05 ^a^	0.92 ± 0.14 ^ab^	1.11 ± 0.1 ^bc^	1.28 ± 0.18 ^c^
HPI	1.04 ± 0.14 ^abc^	1.16 ± 0.09 ^c^	0.86 ± 0.04 ^a^	0.88 ± 0.13 ^ab^	1.08 ± 0.0 ^bc^	1.24 ± 0.16 ^c^
UI	186 ± 11.7 ^c^	179 ± 17.2 ^bc^	150 ± 6.5 ^a^	163 ± 14.8 ^ab^	176 ± 7.5 ^bc^	173 ± 12.8 ^bc^
FLQ	24.6 ± 2.4 ^c^	22.7 ± 1.8 ^bc^	18.4 ± 1.6 ^a^	20.8 ± 2.3 ^ab^	21.5 ± 1.2 ^abc^	20.8 ± 2.0 ^ab^
PI	0.88 ± 0.06 ^c^	0.84 ± 0.07 ^c^	0.58 ± 0.06 ^a^	0.68 ± 0.108 ^ab^	0.80 ± 0.08 ^bc^	0.82 ± 0.06 ^c^

Values are means of three separate replicates. Means within the same row without a common lowercase letter differ significantly (*p* < 0.05). EPA, eicosapentaenoic acid; DHA, docosahexaenoic acid; SFA, saturated fatty acids; PUFA, polyunsaturated fatty acids; UNS, unsaturated fatty; ARA, arachidonic acid; AI, atherogenic index; TI, thrombogenicity, h/H, hypocholesterolaemic/hypercholesterolaemic fatty acid ratio, HPI, health-promoting index; UI, unsaturation index; FLQ, flesh lipid quality; PI, polyene index.

**Table 4 foods-12-01595-t004:** EPA + DHA contents (mg/100 g wet weight) in control group (CTR) and glyphosate-exposed group (Gly) at 20 and 26 °C. Values are means ± standard deviations.

T°C	Gly	EPA +DHA (mg/100 g ww)	Mussel Portion (g) Containing Recommended EPA + DHA	
			250 mg	500 mg
20 °C	CTR	227 ± 23.7 ^c^	110	220
	1 mg/L	182 ± 13.1 ^b^	137	275
	10 mg/L	150 ± 12.9 ^a^	167	334
26 °C	CTR	200 ± 15.6 ^bc^	125	250
	1 mg/L	211 ± 8.6 ^bc^	119	238
	10 mg/L	173 ± 17.3 ^b^	144	288

Means in a column with different letters are significantly different (*p* < 0.05). EPA = eicosapentaenoic acid; DHA docosahexaenoic acid.

## Data Availability

Data is contained within the article.

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
