# Peer review of "Can Glyphosate and Temperature Affect the Nutritional Lipid Quality in the Mussel Mytilus galloprovincialis?"

_foods, 2023, doi:10.3390/foods12081595_

Round 1

Reviewer 1 Report

This is an interesting manuscript. Please see the following comments. Please in the revision provide a seperate sheet where you answer each comment in full and that you show the amendments marking the lines in the revised track changes or non track changes version of the manuscript

1) some moderate English language corrections are needed

see some of the following

replace

a critical role in human health through protection against a variety of diseases

with

a critical role in human health against a variety of diseases

replace

In general, statistical analysis (Two-way Anova) showed significant effect of

with

Statistical analysis  showed a significant effect of

replace

leading a decline of nutritive 19 values and quality of mussels

with

leading to a decline of nutritive 19 values and to the quality of mussels

you state

Marine ecosystems occupy an optimal position in meeting human nutrition needs as 26 are a rich source of nutritious food.

unclear meaning please rephrase 

replace

 for both its organoleptic, competitive price and nutritive prop- 31 erties

with

 for both their organoleptic and nutritional properties and their competitive price 

replace

all the important metabolic functions that these fatty acids possess

with

all the important metabolic functions that these fatty acids facilitate

replace

 due to the massive use of pesticid

with

 due to the massive use of plant protection products

replace

Among them rising 94 water temperatures associated with global climate change, represents the main concern 95 for worldwide scientists and politicians

with

Among them rising 94 water temperatures associated with global climate change, represent an important concern 95 for  scientists and regulatory bodies worldwide 

you state

 on lipid content ww revealed

there is something wrong here

you state

Lipid Nutritional Quality Indexes of M. galloprovincialis. 

it should be indices and also M. galloprovincialis should be in italics

there are more mistakes please proofread throughout

2) for all materials and methods first of all give a separate subchapter showing the supplier (name, city and country of origin) for all reagents and chemicals that you used. also for all apparatuses used eg oven, filters, carrier gases, etc give manufacturer, city and country of origin

3) it is not clear to me when you describe the indices-do you have the data for eg C12: 0, C14: 0 etc... are these concentrations in what units? were they provided by the analytical procedure in 2.2 because you never really described what you measured on GC. Also the sum of PUFAS etc in these indices how were calculated? in general you have to give more information on the data you show there

4) in 2.4. Statistical analyses the description is very good, thank you for this

5) it is a bit tiring and confusing for the reader to see all the quantitative information of the differences between groups also in the text while they are found in the graphs and tables. please give only a qualitative description of what changes between each exposure groups and do not give actual values when possible. as such the results section should be reduced in length.

6) since it is a two way ANOVA that checks the interaction between the two parameters please explain to me what the post hoc letters a, b, c etc signify-there is a statistically significant difference between what and what

7) if possible to cite the following papers on mediterranean mussels and pollution

Chemosphere Volume 119, Pages S145 - S152 2015

 Marine Environmental Research, 1999 47 (5), pp. 415-439

Environment International, 2009 35 pages 599-606.

8) one important drawback of the study is that glyphosate was not actually measured in the in vivo exposure if I am correct. how you were sure that the nominal concentration was the actual concentration? please clarify

Author Response

Response to Reviewers.

We thank review  for thoughtful suggestions and insights, which have enriched the manuscript and produced a better and more focused report on the research topic.

You can find the changes required by reviewer 1 in  text in red.

Reviewer #1:

This is an interesting manuscript. Please see the following comments. Please in the revision provide a seperate sheet where you answer each comment in full and that you show the amendments marking the lines in the revised track changes or non track changes version of the manuscript

1) some moderate English language corrections are needed see some of the following

Review Comment: replace a critical role in human health through protection against a variety of diseases with a critical role in human health against a variety of diseases

Authors Response: it has been done

Review Comment: replace In general, statistical analysis (Two-way Anova) showed significant effect of with Statistical analysis showed a significant effect of

Authors Response: it has been done

Review Comment: replace leading a decline of nutritive values and quality of mussels with leading to a decline of nutritive values and to the quality of mussels

Authors Response: it has been done

Review Comment: you state Marine ecosystems occupy an optimal position in meeting human nutrition needs as are a rich source of nutritious food. unclear meaning please rephrase 

Authors Response: the sentence has been improved and clarified

Review Comment: replace for both its organoleptic, competitive price and nutritive properties With for both their organoleptic and nutritional properties and their competitive price 

 Authors Response: it has been done

Review Comment: replace all the important metabolic functions that these fatty acids possess

With all the important metabolic functions that these fatty acids facilitate

 Authors Response: it has been done

 Review Comment: replace due to the massive use of pesticide with  due to the massive use of plant protection products

 Authors Response: it has been done

Review Comment:  replace Among them rising water temperatures associated with global climate change, represents the main concern for worldwide scientists and politicians with Among them rising water temperatures associated with global climate change, represent an important concern for scientists and regulatory bodies worldwide 

 Authors Response: it has been done

Review Comment: you state on lipid content ww revealed  there is something wrong here

Authors Response: Thank You for the observation. ww has been deleted. 

Review Comment: you state Lipid Nutritional Quality Indexes of M. galloprovincialis. 

it should be indices and also M. galloprovincialis should be in italics there are more mistakes please proofread throughout

Authors Response: Thank You for the observation, we made all the changes

 Review Comment: 2) for all materials and methods first of all give a separate subchapter showing the supplier (name, city and country of origin) for all reagents and chemicals that you used. also for all apparatuses used eg oven, filters, carrier gases, etc give manufacturer, city and country of origin+

Authors Response: We have followed the suggestions and inserted a separate subchapter with the requested information

 Review Comment:  3) it is not clear to me when you describe the indices-do you have the data for eg C12: 0, C14: 0 etc... are these concentrations in what units? were they provided by the analytical procedure in 2.2 because you never really described what you measured on GC. Also the sum of PUFAS etc in these indices how were calculated? in general you have to give more information on the data you show there

Authors Response: We have calculated the relative quantity of the fatty acids that were expressed as percentage of total fatty acids. Moreover, we have calculated the absolute quantity (mg/100g ww) of fatty acids by using the internal standard. However, for the calculation of indices, since the most of them are ratios, the value does not change regardless of the unit of measurement used.

Review Comment:  4)  in 2.4. Statistical analyses the description is very good, thank you for this

Authors Response:  Thank you for your agreeable comments.

Review Comment:  5) it is a bit tiring and confusing for the reader to see all the quantitative information of the differences between groups also in the text while they are found in the graphs and tables. please give only a qualitative description of what changes between each exposure groups and do not give actual values when possible. as such the results section should be reduced in length.

Authors Response: We agree with your comment, so we have delayed the quantitative values.

Review Comment: 6) since it is a two way ANOVA that checks the interaction between the two parameters please explain to me what the post hoc letters a, b, c etc signify-there is a statistically significant difference between what and what

 Authors Response: We have always  used the post-hoc tests because many times interaction is non-significant but post-hoc tests show that some comparisons reach significant levels (See our MUFA results). This post-hoc includes the comparison among all treatments: between the two temperatures tested and among  glyphosate treatments.

Review Comment:7) if possible to cite the following papers on mediterranean mussels and pollution

Chemosphere Volume 119, Pages S145 - S152 2015

 Marine Environmental Research, 1999 47 (5), pp. 415-439

Environment International, 2009 35 pages 599-606.

 Authors Response: The suggested papers have been cited

Review Comment:8) one important drawback of the study is that glyphosate was not actually measured in the in vivo exposure if I am correct. how you were sure that the nominal concentration was the actual concentration? please clarify

 Authors Response: We have verified the nominal glyphosate concentrations of the experimental solutions using HPLC, since the differences between nominal and measured concentrations were negligible, we have decided of not include them in the text.

Reviewer 2 Report

This study aimed at evaluating the combined effect of glyphosate (Gly), one the most commonly used pesticides worldwide, and temperature (T°C) on the quality of lipid and fatty acids in the mussel Mytilus galloprovincialis. The topic is interesting, but a similar study on the effects of temperature and Gly on FAs composition of the gastropod Lymneae sp has been already conducted, so the originality of the present paper is not high. The previous paper is cited and results are compared. Also, the other species is examined,. Material and methods are adequately described, results are clearly presented. There are many references cited in the paper, and it is especially commendable that many papers that have been published recently are cited.

Author Response

Reviewer #2:

This study aimed at evaluating the combined effect of glyphosate (Gly), one the most commonly used pesticides worldwide, and temperature (T°C) on the quality of lipid and fatty acids in the mussel Mytilus galloprovincialis. The topic is interesting, but a similar study on the effects of temperature and Gly on FAs composition of the gastropod Lymneae sp has been already conducted, so the originality of the present paper is not high. The previous paper is cited and results are compared. Also, the other species is examined,. Material and methods are adequately described, results are clearly presented. There are many references cited in the paper, and it is especially commendable that many papers that have been published recently are cited.

Authors Response:  Thank you for your agreeable comments.

Reviewer 3 Report

This study determine the lipid quality (fatty acid content and profiles as well as LNQI) of mussel Mytilus galloprovincialis. The topic is interesting. If the experimental designs and discussion is well-presented, some new findings will get from this study.

However, in my opinion, this MS still required major revision. Actually, the determination on lipid quality/experiments is very simple (without any up-to-date assays), only new sample and determination of LNQI are novel which should be highlighted in this MS. Thus, the introduction should straight out this point. Now, the research was conducted at a reasonable level, but lack of in-depth exploration. Abstract should be reconstructed. Results should be adjusted for easier reading. Also, discussion should be precise and not too general (should focus on the effect of gly and temperature). Here is my comments:

ABSTRACT

- Line9-12: The first introduced sentence in the abstract is too general. I suggested author state or give the details of how much omega-3-PUFAs are contained in the mussel Mytilus galloprovincialis instead of seafood.

- Line13: What’s temperature i.e. temperature for culturing/ for storage/ etc.? Please identified. Moreover, I do not understand the meaning of “(T℃)”, please check.

- Line14: How’s different between the quality of lipids and fatty acids, particularly fatty acids? At this sentence, I suggested to use “on quality of lipids including…(parameters)………….”

- Line21: 26 or -26℃????

- Overall, the abstract should be totally revised/improved. The structure/order should be reconstructed. Abstract should start with how’s important/introduction of this study (line9-12). Then, followed by the objectives or scope of this study. Moreover, some empirical results, particularly quantitative data should more state to strengthen the abstract, with statistical analysis. Finally, the findings obtained from this study should clearly provide. The idea for further use/further studies will then follow.   

INTRODUCTION

- Line 29,34, 59, 66, etc. : No need to start the new paragraph

- Line40: Please give the ref. I suggested author to add information how much PUFA is contained in this mussel species. This information can be included in the abstract to strengthen how’s important of this study.

- Most sentences/ information is still lack of ref. like line40, 48. Please check of the appropratie to bring the data/theory.

- The main/important data which should appear in the introduction is still lacking. First, for the Glyphosate (Gly), what is the hypothesis that this factor can affect to lipid quality? Are there any previous studies exhibited on it affected to lipid quality? Next, what is the temperature used in this study? I mean storage or culturing temperature or what? And what is the hypothesis behind this factor? Also, If it had some previous studies to support, it would be great. Finally, what is the combined effects (line105)? Why this study aimed to evaluate the combined affect since it looks like no information related to each factor in the past?

- Moreover, the introduction is too general. Author give basically information related to lipids, PUFA, omega oil, which everybody already know. Author should give more previse information related to this study like “LNQI” is interesting, etc.

MATERIAL AND METHODS

- Line112: What is the specification of adults? I suggested providing more information related to raw materials likes weight, period of culture/purchase, etc.

-Line113: Why author should give the ref. here? Is this ref. paper claimed that samples were captured from unpolluted area?

- It seem like author start the experiment with the live animal. It means that there will have the slaughtering procedure, thus, how’s about the ethic approval for this study?

- Line116: Acclimate into…. Please clarify.

- Line116: exposed to Gly at concentration of………… Please clarify.

- Line126: Why this study selects 20 and 26℃ ??? How author control the temperature during the process.

RESULTS AND DISCUSSION

- All discussion should improve. The conclusion or findings from the results should be re-conclude for precise information. At this point, the discussion is too general. For the results and discussion, there is a lack of depth and logic. And, the conclusion need to be rewritten.

- Figures and tables should appear with the text at the appropriate positions (when first discuss).

- Figure1, particularly %moisture, the ratio of graph is not appropriate. All samples contained moisture content around 85-90%, which some are significantly difference, but the ratio cannot see the different. Please adjust.

Author Response

We thank review  for thoughtful suggestions and insights, which have enriched the manuscript and produced a better and more focused report on the research topic.

You can find the changes required by reviewer 3 in  text in Blue;

Reviewer #3:

This study determine the lipid quality (fatty acid content and profiles as well as LNQI) of mussel Mytilus galloprovincialis. The topic is interesting. If the experimental designs and discussion is well-presented, some new findings will get from this study.

Authors Response: Thank you for your agreeable comments.

Review Comment: However, in my opinion, this MS still required major revision. Actually, the determination on lipid quality/experiments is very simple (without any up-to-date assays), only new sample and determination of LNQI are novel which should be highlighted in this MS. Thus, the introduction should straight out this point. Now, the research was conducted at a reasonable level, but lack of in-depth exploration. Abstract should be reconstructed. Results should be adjusted for easier reading. Also, discussion should be precise and not too general (should focus on the effect of gly and temperature).

Authors Response: We thank you for your thoughtful suggestions and insights, which have enriched the manuscript and produced a better and more focused report on the research topic.

Here is my comments:

ABSTRACT

Review Comment: - Line9-12: The first introduced sentence in the abstract is too general. I suggested author state or give the details of how much omega-3-PUFAs are contained in the mussel Mytilus galloprovincialis instead of seafood.

Authors Response: We have revised the Abstract according to your comment.

Review Comment: - Line13: What’s temperature i.e. temperature for culturing/ for storage/ etc.? Please identified. Moreover, I do not understand the meaning of “(T℃)”, please check.

Authors Response: T℃ is referred to temperature on the Celsius scale. So, degrees
Celsius. We referred to culturing temperature

Review Comment: - Line14: How’s different between the quality of lipids and fatty acids, particularly fatty acids? At this sentence, I suggested to use “on quality of lipids including…(parameters)”

Authors Response: We agree with Review and revised the sentence

Review Comment: - Line21: 26 or -26℃????

Authors Response: We apologize for the lack of measurement units

Review Comment: - Overall, the abstract should be totally revised/improved. The structure/order should be reconstructed. Abstract should start with how’s important/introduction of this study (line9-12). Then, followed by the objectives or scope of this study. Moreover, some empirical results, particularly quantitative data should more state to strengthen the abstract, with statistical analysis. Finally, the findings obtained from this study should clearly provide. The idea for further use/further studies will then follow.    

 Authors Response: Based on Your valuable comment, we have revised and improved the abstract.

INTRODUCTION

Review Comment:- Line 29,34, 59, 66, etc. : No need to start the new paragraph

 Authors Response:: Thank You for the right observation, we didn't decide it, but the Foods Editorial Office

Review Comment:-  Line40: Please give the ref. I suggested author to add information how much PUFA is contained in this mussel species. This information can be included in the abstract to strengthen how’s important of this study.

Authors Response:: We thank You for your useful suggestion and we added information on PUFAs content in mussel.

Review Comment:-   Most sentences/ information is still lack of ref. like line40, 48. Please check of the appropratie to bring the data/theory.

Authors Response: We agree with your comment and the references have been added

Review Comment:-   The main/important data which should appear in the introduction is still lacking. First, for the Glyphosate (Gly), what is the hypothesis that this factor can affect to lipid quality? Are there any previous studies exhibited on it affected to lipid quality? Next, what is the temperature used in this study? I mean storage or culturing temperature or what? And what is the hypothesis behind this factor? Also, If it had some previous studies to support, it would be great. Finally, what is the combined effects (line105)? Why this study aimed to evaluate the combined affect since it looks like no information related to each factor in the past?

Authors Response: We apologize for not providing detailed information. We have reported previous studies on the effect of temperature and glyphosate as individual stressors on lipid and fatty acids profile. Therefore, we have added the requested information.

Review Comment:- Moreover, the introduction is too general. Author give basically information related to lipids, PUFA, omega oil, which everybody already know. Author should give more previse information related to this study like “LNQI” is interesting, etc.

 Authors Response: We agree with Review and added the requested information

MATERIAL AND METHODS

Review Comment:- - Line112: What is the specification of adults? I suggested providing more information related to raw materials likes weight, period of culture/purchase, etc.

Authors Response: We agree with Review and added the requested information

Review Comment:- -Line113: Why author should give the ref. here? Is this ref. paper claimed that samples were captured from unpolluted area?

Authors Response: the reference does not state that the samples were taken in an unpolluted area, but that that area is unpolluted and our samples come from there

Review Comment:- It seem like author start the experiment with the live animal. It means that there will have the slaughtering procedure, thus, how’s about the ethic approval for this study?

Authors Response: For the experiment of exposure to glyphosate and temperature we have used live animal. No ethical approval is required for invertebrates.

Review Comment:- Line116: Acclimate into…. Please clarify.

Authors Response: We have clarified

Review Comment:- Line116: exposed to Gly at concentration of………… Please clarify.

Authors Response: We have clarified

Review Comment:- Line126: Why this study selects 20 and 26℃ ??? How author control the temperature during the process.

Authors Response: the temperatures were chosen based on the temperature range recorded in the mussel farm area. The experiments were conducted in thermostated chambers under controlled temperature.

RESULTS AND DISCUSSION

Review Comment:- - All discussion should improve. The conclusion or findings from the results should be re-conclude for precise information. At this point, the discussion is too general. For the results and discussion, there is a lack of depth and logic. And, the conclusion need to be rewritten.

Authors Response: According to Review comments, and following the useful suggestions we have improved the text.

Review Comment:- - Figures and tables should appear with the text at the appropriate positions (when first discuss).

Authors Response: Thank You for the right observation, we didn't decide it, but the Foods Editorial Office

Review Comment:- - Figure1, particularly %moisture, the ratio of graph is not appropriate. All samples contained moisture content around 85-90%, which some are significantly difference, but the ratio cannot see the different. Please adjust.

Authors Response: According to Review suggestion, we modified Figure 1

Round 2

Reviewer 3 Report

The authors conducted the required revisions as suggested before.

There are a few comments related to the abstract as follows:

- Line19: and?

- Line19-20: Some FAs among all group did not significantly different, thus I suggested author to add (or give dominant FA) which showed significant difference among groups.

- I suggested to end up the abstract with the idea for further use/further studies or the application of these findings.

Author Response

Thank you for useful suggestion you can find the change (in red) in  the text 

  • Line19: and?
  • Author replay: yes , we have corrected
  •  
  • Line19-20: Some FAs among all group did not significantly different, thus I suggested author to add (or give dominant FA) which showed significant difference among groups.
  • Author replay: We have added information data.
  •  
  • I suggested to end up the abstract with the idea for further use/further studies or the application of these findings 
  • Author replay: We have added a sentence  at the end of the abstract, about further study on this focus.